# How Were Return-of-Service Schemes Developed and Implemented in Botswana, Eswatini and Lesotho?

**DOI:** 10.3390/healthcare11101512

**Published:** 2023-05-22

**Authors:** Sikhumbuzo A. Mabunda, Andrea Durbach, Wezile W. Chitha, Oduetse Moaletsane, Blake Angell, Rohina Joshi

**Affiliations:** 1School of Population Health, University of New South Wales, Sydney, NSW 2052, Australia; 2The George Institute for Global Health, University of New South Wales, Sydney, NSW 2042, Australia; 3Australian Human Rights Institute, University of New South Wales, Sydney, NSW 2052, Australia; 4Health Systems Enablement and Innovation Unit, University of the Witwatersrand, Johannesburg 2000, South Africa; 5Pharmacovigilance and Clinical Trials, Botswana Medicines Regulatory Authority, Gaborone P.O. Box 505155, Botswana; 6The George Institute for Global Health India, New Delhi 110025, India

**Keywords:** health system, health workforce, human resources, health policy, bursary

## Abstract

Botswana, Eswatini and Lesotho are three Southern African countries that make use of return-of-service (RoS) schemes to increase human resources for health in their countries. These initiatives bind beneficiaries to a pre-defined period of service upon the completion of their studies based on the length of funding support received. We aimed to review the history of these policies to understand the conceptualisation, intent and implementation of these schemes. We used a multi-methods research design which consisted of a literature review, a policy review and semi-structured interviews with policymakers and implementors. All three governments have a combination of grant-loan schemes and full bursaries or scholarships. The policies have all been operating for over 20 years, with Eswatini’s pre-service policy being the oldest since it was introduced in 1977, followed by Lesotho’s 1978 policy and Botswana’s 1995 pre-service policy. These policies have never been reviewed or updated. RoS schemes in these countries were introduced to address critical skills shortages, to improve employability prospects for citizens, to have competent public sector employees by global standards and to aid the career progress of government employees. Ministries of Health are passive role players. However, these schemes can only be efficient if there is clear cooperation and coordination between all stakeholders.

## 1. Introduction

With high burdens of morbidity and premature mortality, many Southern African countries are failing to meet their health targets [1]. Contributing to this are the marked human resources for health (HRH) shortages and maldistribution [1,2]. Botswana, Eswatini (formerly Swaziland) and Lesotho are three landlocked middle-income Southern African countries (Table 1), which are mostly rural and have pronounced health worker shortages, especially in rural areas [3,4,5,6,7,8,9]. With Lesotho completely surrounded by South Africa, all three countries share borders with South Africa [7,8,9]. In 2019, these three countries had the highest reproductive-age (15–49 years) HIV prevalence in the world, with Eswatini ranking first and Botswana ranking third [10]. Lesotho has the highest tuberculosis (TB) incidence in the world, which stood at 650/100,000 people in 2020 [11]. All three countries have a doctor-to-population ratio of below 5 per 10,000 population and a nurse and midwife population ratio of below 42 per 10,000 population [12]. The ratio of doctors, nurses and midwives for each of the three countries is therefore below the United Nation’s Sustainable Development Goal target of 44.5 per 10,000 population [1,13,14]. 

These shortages are associated with the limited training capacity in these countries, the emigration of health professionals to other countries, the dual health systems and growing population sizes [1,3,4,5,17]. Unique to Botswana is the fact that only 16% of the country is inhabitable, as 84% is dominated by the Kalahari Desert, 60% of the population is concentrated in the Eastern part of the country and 40% live in rural areas, most of which are sparsely populated, further complicating the provision of health services [18]. These countries use return-of-service (RoS) scheme training strategies to increase the production of new entry health professionals, nursing specialists and medical specialists [4,18,19,20,21]. Through RoS schemes, governments fund aspiring health professionals or health professionals in employment to respectively study towards a health profession or specialise in return for service [14,21,22]. Of the three countries, Botswana is the only one that has had a medical school since 2009, creating some internal training capacity for medical students and specialists [17,18,21,23]. Notwithstanding, all these countries send beneficiaries to neighbouring countries and abroad for their studies [18,19,20]. There is no literature outlining the policy intentions, planning process and recruitment into service of the beneficiaries, casting doubt on the effectiveness and value for money of these schemes. These aspects can only be understood once the histories, evolution and architecture of these policies are understood. This study, therefore, aimed to describe the design and implementation of RoS schemes in Botswana, Eswatini and Lesotho and further understand how they adhere to the principles of the World Health Organization (WHO) framework for human resources for health (HRH) development in Africa [24,25].

## 2. Materials and Methods

### 2.1. Design

Data were derived through a multimethod research design including: (a) a literature review, (b) a policy review and (c) semi-structured interviews of policymakers and implementors. The Consolidated criteria for Reporting Qualitative research (COREQ) and the Checklist for the use and Reporting of Document Analysis (CARDA) were used for the article [26,27].

### 2.2. Recruitment and Data Collection

The aim of the literature review was to review information on the history of bursary or scholarship schemes with an expectation of a government return-of-service. First, a literature search was undertaken through searching databases (Pubmed, CENTRAL and EBSCO Host (including CINAHL)) between 1 October 2020 and 31 August 2022. Next, the archives of the National Libraries in Gaborone, Mbabane and Maseru were searched. Databases were searched using the search terms: (‘bursar*’ OR ‘scholarship*’ OR ‘return of service’ OR ‘return for service’ OR ‘in-service’ OR ‘grant*’ OR ‘financial aid’) AND (‘health’) AND (‘Eswatini OR Swaziland’ OR ‘Botswana’ OR ‘Lesotho’) to identify relevant literature examining the history of RoS schemes, studies of their implementation or operation and their evolution over time. Data were extracted using a designed and piloted tool to extract relevant information on the RoS policy context, content, process and actors (Appendix A). Second, we contacted individuals with ultimate responsibility for the operation of the schemes in the three countries, specifically: the Director of Tertiary Education Financing (Ministry of Education) and the Director of Human Resource Development (Ministry of Health) in Botswana; the Permanent Secretaries in the Ministries of Labour and Public Service in Eswatini; the Director of the National Manpower Development Secretariat (NMDS) and the Deputy Director of Human Resource Development (Ministry of Health) in Lesotho. These senior officers were requested for access to officials who were the most relevant to be interviewed on RoS schemes for health professionals. A total of nine policymakers were identified, with two in Eswatini, three in Lesotho and four in Botswana. Participants were excluded if they had less than 2 years of experience in the role or refused to consent. Microsoft Teams virtual platform was used for interviews to gain a deeper understanding of how RoS policies were designed and implemented and how they have evolved over time (Appendix A). Participants were interviewed as individuals (four) or in the presence of other colleagues (five). As part of the interview process, they were allowed to refer to policy documents to confirm certain aspects. The interviewer took field notes along with audio recordings of the interviews using Microsoft Teams. Interviews took an average of 64 min. Transcripts were obtained from Microsoft Teams, transferred to Microsoft Word and cleaned by the first author, who also translated the non-English language phrases (siSwati, seSotho and seTswana) used in the interviews. Lastly, between 1 November 2020 and 15 February 2022, policymakers were requested to provide all existing versions of policy documents that were on record since the inception of the schemes for the policy review. We requested all RoS scheme-related documents including contract templates, RoS scheme beneficiary selection criteria, correspondents templates with stakeholders and beneficiaries, audit reports, etc. Similar to the literature review, this policy review extracted information on the piloted data extraction tool (Appendix A).

### 2.3. Reflexivity

The first author (S.A.M.) is a South African Public Health Medicine Specialist who held several leadership positions as an undergraduate medical student at the University of Cape Town in the early 2000s. This exposed him to agreements that the university had with other governments. In those years, he also studied with and befriended RoS beneficiaries from a number of Southern African countries, including the three countries of interest in this study.

### 2.4. Data Analysis

The Walt and Gilson policy analysis framework [28] was used to analyse both the policy document review and literature review and were summarised using a narrative approach to reporting. An approach of familiarisation, coding, theme development, review of themes, theme definition and reporting was followed for the qualitative interview analysis. The qualitative interview results are presented based on four main themes and nine sub-themes that were derived using inductive thematic analysis in NVIVO 12. There was member-checking of transcripts by participants to verify the accuracy of the information. Peer-checking of half of the transcripts by B.A. and R.J. helped devise the themes, and all authors reached consensus on the final themes.

### 2.5. Ethics Approval

Ethics approvals were obtained from the University of New South Wales (HC200519), the Botswana Health Research and Development Division (HPDME 13/18/1), the Botswana Ministry of Tertiary Education, Research, Science and Technology (DRST 7/2/13 XXVII (27)), the Eswatini Health and Human Research Review Board (FWA 00026661/IRB 000 11253) and the Lesotho Ministry of Health Review Board (ID 24-2021). Participants have been de-identified. Direct quotes are labelled using random labels known only by the first author (S.A.M.) and senior authors (B.A. and R.J.).

## 3. Results

The literature review identified a total of 668 records, of which 45 were deemed as potentially suitable for our review. Following the full-text appraisal of these texts, 21 were included in the final review (Figure 1). All approached policymakers agreed to interview. The interview respondents were four officials from Botswana (three from the Ministry of Health and one from the Ministry of Education), two from Eswatini (one from the Ministry of Labour and one from the Ministry of Public Service) and three from Lesotho (two from NMDS and one from the Ministry of Health). Of these participants, four were female and five were male, and they had experience that ranged from 5 to 22 years in administering the schemes. There was good alignment between the three data sources, as such information is presented under a single theme, marked by citations if extracted from the literature or policies; direct statements are in italic font.

### 3.1. Understanding RoS Schemes in Three Southern African Countries (Context and Content)

Using the Walt and Gilson policy analysis framework [28], Figure 2 was produced.

#### 3.1.1. Types of Schemes

Broadly, the governments have two programmes that benefit health workers, namely, the in-service programme for government employees and the pre-service programme for future employees, mostly youth with high academic grades at school [29,30,31,32,33,34] (Table 2). 

The stewardship for the in-service programmes is the Ministry of Health in Botswana, the Ministry of Public Service in Eswatini and the NMDS in Lesotho [30,31,32,33,34]. Whilst the NMDS is also responsible for the pre-service programme in Lesotho, the Ministry of Education and Ministry of Labour are responsible for the pre-service programme in Botswana and Eswatini, respectively [4,17,18,19,20,32,33,35,36,37,38]. 

Botswana has two main schemes under the pre-service programme, namely, the top achievers’ programme (TAP) and the grant-loan scheme [38]. Whereas the grant-loan scheme only funds beneficiaries for undergraduate studies, the TAP starts funding a beneficiary from undergraduate pre-entry studies (e.g., A-levels, O-levels, pre-medical studies, etc.) to doctoral studies. The grant-loan scheme uses a sliding scale of four categories that prioritises beneficiary groups per academic programme of study, with health sciences, engineering and all skills deemed to be scarce being ranked at the top, and the last category is performing arts, which is regarded as an optional course.

#### 3.1.2. Aim of the Programmes

RoS schemes were suggested to serve six main aims whose purpose is to build the health workforce capacity, strengthen the economy and improve educational opportunities for citizens. 

(a)Address critical skills shortages and strengthen government capacity

Countries often reflect on health workforce shortages, especially nursing and medical specialists. This is due to the recognition that the training of employees in specialist fields helps reduce specialist referrals to neighbouring countries (e.g., South Africa) and the private sector. 

GBL1: *For example, we refer to South Africa, we refer to private hospitals, we refer to medical laboratories… So, for us it’s an issue of not having the right skilled manpower. …you don’t have to refer for cancer or something like that, …we can do on our own home ground. So, that within four or five years, uh, let me say ten years at least, we are able to be sustainable. We don’t have to refer people for cardiology…*

One of the principal purposes of Eswatini’s Ministry of Labour is to manage issues of the supply and demand of human resources into the labour market. 

PP2: *So, I study the …dynamics in the labour market, and then say for the labour market to be efficient and effective, what does it require? Obviously, one of the major… inputs into it, it’s human resources. … that’s where then we have to then have very clear human resource planning and development strategies, …skills, uh, requirements in the short term…, in the medium term, in the long-term…*


(b)Professionals who are relevant and up-to date

It is important for the public service to have employees who have kept up with the latest knowledge trends in their profession.

PP1: *I guess the objectives would be for government to gain because we live in an ever-changing world, so, we need to constantly have skills that are…, on par with the rest of the world (sic).*

(c)Human resource development and career pathing

These schemes are used for the development and career progression of existing health employees. This allows employees to benefit from global academic opportunities and thus further benefit their countries. For instance, midlevel health workers e.g., pharmacy assistants and dental therapists, might be offered opportunities to be pharmacists and dentists, respectively. Medical doctors may specialise in fields such as psychiatry, obstetrics and gynaecology, internal medicine, emergency medicine, pathology, etc. Nurses, dentists and allied health workers can also specialise further, thus increasing their individual earning potential whilst contributing to the health system.

(d)Improve employability prospects for citizens

At least one of each of the countries’ policies places emphasis on the need to ensure that citizens are prioritised to take up job opportunities in all sectors, especially the public sector [18,29,30,31,32,33,38,39,40,41]. Foreigners are only employed when there is no citizen who can perform the same job. In that event, the foreigner and/or employer needs to have plans in place to ensure that there is skill transfer to citizens for when the foreigners leave. As supported by a participant:

PP2: *So, at independence you know the country had a desire to fill, uh, strategic positions of the economy with, uh, qualified emaSwati [Swati nationals]. So, uhm, there was then a targeted, uhm programme to train emaSwati [Swati nationals] so that when they come back, they will take up their positions, and the policy then was called local, it was called a localisation policy.*


(e)Strengthen management and population health skills

In-service programmes also help to enhance identified management gaps in all the ministries in Eswatini and Lesotho and in only the Ministry of Health in Botswana. Population health skills are also strengthened through these programmes.

(f)Fulfill national political mandates

The three countries receive guidance for training health professionals from the governments’ macro policy instruments such as the Poverty Reduction Strategy and the National Development Strategy (NDS) in Eswatini [37,42], Vision 2016 [43] in Botswana and Lesotho’s National Strategic Development Plan (NSDP) [35]. The NDS, Vision 2016 and NSDP place access to health in the highest level of priority [35,37,42,43]. These documents are often written by consultants. As narrated by a participant: 

PP2: *So, this was like the start…, the span of this study was like 5 years. So, uh, that priority, identification came to an end this past year, 2020. So, we have since appointed a consultant now to undertake a new study, you know, to project… the training needs for the next five years again.*

#### 3.1.3. Who Are the Beneficiaries?

Table 3 summarises the characteristics of the beneficiaries, and it shows that being a citizen is common for all the policies and countries. Pre-service schemes place lots of weight on academic merit. Individuals applying for in-service schemes should apply for a programme that is in line with their current job. Whilst the criteria used to select beneficiaries are generally transparent, however, this is not always the case in Lesotho. These are set standard criteria and do not vary from year to year.

DS1: *…we don’t want students knowing our criteria because sometimes, we try to share the criteria with them… But you know how students are, if you didn’t succeed, now… they go back to criteria. Now I passed more than whoever blah blah blah.*


Even though Botswana’s in-service programme is used to exclude pregnant employees for academic sponsorship, that has since changed, as narrated by a participant.

GBL1: *We are trying to cater for everybody. You know? So, pregnancy is not no longer an issue.*


#### 3.1.4. What Are the Beneficiary Obligations?

Beneficiaries are contracted/bonded to return for service after the completion of their studies, as per Table 3.

Further shown in Table 3 is that undergraduate health sciences beneficiaries in Botswana are required to serve a one-year service period for each year of funding support at the end of their studies. Botswana’s grant-loan scheme beneficiaries, Eswatini’s Ministry of labour beneficiaries and Lesotho’s NMDS beneficiaries are also not compelled to serve in the government if they are within the country. However, whilst those who serve in the public sector in Lesotho must repay 50% of the funding support within five years of the completion of their studies, those who work in the public sector have to repay 65%. 

Lesotho’s NMDS beneficiaries serve for a minimum of three years if funded for a year. Beyond a year, it is twice the service period for each year of funding. The latter condition is similar for Eswatini’s pre-service schemes. Eswatini’s pre-service scheme (Ministry of Labour’s) beneficiaries must repay 50% of the funding support if in the country and 100% if outside the country. In-service beneficiaries from Eswatini’s Ministry of Public Service scholarships are mandated to serve a year more than the funded period, something similar for Botswana’s Ministry of Health’s in-service beneficiaries funded for more than a year. The maximum period of service is 5 years for the latter, regardless of the funding duration. 

PP1: *Officers who are sent on training are expected back in the positions they were prior to leaving. We do not allow for officers to change professions under this facility.*

#### 3.1.5. Are There Possibilities for Contract Deviations?

If a Ministry of Labour-funded beneficiary in Eswatini fails an academic year, then the funding is suspended for the following year and completely cancelled after two consecutive years of non-progression. If the beneficiary is the one who voluntarily opts out of the academic programme, then they are obliged to repay 100% of the funding support received. Botswana’s Ministry of Education applies a similar principle but also offers beneficiaries an opportunity to complete their studies at the University of Botswana “where it is much cheaper”. 

If an in-service beneficiary in Botswana wants to change profession (e.g., nurse to medical doctor), then they must address a motivational letter to the responsible board, who will then deliberate and decide on the matter. A similar process is also followed for medical specialists who could decide to sub-specialise (2 years) after the completion of their initial 4-to-5-year specialist training (e.g., internal medicine specialty to nephrology, etc.). 

GBL1: *I think that one is discretion, the Board will decide whether you sign another bond, because this one would have been five years and then you sign another one which is for a different programme that becomes additional two years. …it’s normally, uh, a special dispensation because we don’t always allow for somebody to continue, uh, immediately after completing. You must come back, serve a certain period of time then continue with another qualification.*

Whilst Lesotho NMDS beneficiaries are generally not allowed to change academic programmes, exceptions can be made if the new programme is a priority programme for the country in that year. There are also situations when beneficiaries cannot serve their obligatory service period due to a lack of jobs. The important thing is for such beneficiaries to inform the funding Ministry (or NMDS if in Lesotho).

#### 3.1.6. Origins of the Schemes

Anticipating independence (which eventually came in 1968), the Eswatini government launched the localisation and training policy in 1965, where nationals would be trained to take up these strategic positions upon their return [31,39,40]. Building on the localisation policy, in 1977, there was then a formal government scholarship programme which trained nationals for public and private sector positions. Until the year 2000, when Eswatini’s in-service training policy was introduced, it was not structured, as it was guided by the Government’s General orders and training register [31,34]. 

Lesotho’s policy (for both in-service and pre-service beneficiaries) was introduced as the National Manpower Development Council Act in 1978 [32]. This Act is operationalised through the Loan-Bursary Fund Regulations of 1978 [32,33]. The Act legislates the establishment of NMDS, the governance and administration of bursaries and the need to establish regulations governing the loan bursary scheme and determining the rates of allowances payable to existing public sector employees who get awarded bursaries [32,33].

Botswana’s pre-service and in-service policies were launched in 1995 and 1996, respectively [18,29,30,38]. The in-service programme is governed by the 1999 Training Management handbook, which operationalises the relative legislation defining public sector conditions, including conditions for RoS beneficiaries who are offered opportunities as government employees [29,30]. Even though Botswana published guidelines for the award of pre-service schemes in 2011 [38], all five policies quoted in this study have never been reviewed, as expressed by a participant: 

DS1: *…they were introduced in 1978 and they have never been reviewed. We are also, we are only now in the process of reviewing them…*

#### 3.1.7. Policy Development Framework

All three countries’ policy documents are supported by Acts of parliament [29,30,32,33,40,41], and they predate the WHO guidelines for human resources for health in Africa [24,25]. Furthermore, the policies were generic for the whole government and were not limited to health.

#### 3.1.8. Countries of Study

Whilst all three countries have an internal capacity for training non-specialist nurses, midwives, nutritionists and environmental health practitioners, the National University of Lesotho and University of Botswana began training pharmacists in 2003 and 2018, respectively. Botswana is the only country among the three training medical doctors since 2009. Eswatini and Lesotho solely rely on other countries for the training of medical doctors and most other key health professions. Notwithstanding, due to the limited internal training capacity, Botswana still sends medical students and other health sciences beneficiaries to other countries, e.g., United Kingdom, Australia, United States of America, Canada and Cuba being the most common destination countries. South Africa is a popular destination for Botswana’s beneficiaries for all programmes, except for undergraduate medical beneficiaries, who could not be accepted after 2009, when the medical school opened.

GBL2: *…we send…, these employees, around the world to get the gist of these different…, academic fields…, within the different countries that we may have, or we would like them sent to…*

Botswana sends two students to Cuba each year, as a participant elaborates:

FZ1: *So, we have got medical students in Cuba right now. Right now, we have thirteen. We don’t have a lot of students there as much as South Africa has. …Cuba is strictly medicine. But then we also have students who do medicine under the top achievers’ program. This… programme is specifically for high excelling students. …who are at the local institutions doing BSc, Bachelor of Science year one…*


Countries of study are either identified through national government bilateral agreements, donor funding or a select choice of countries and/or universities made by the beneficiary. 

DS1: *So, Lesotho went to the extent of arranging with the government of South Africa for certain institutions to give slots to a maximum of five students per institution for medicine.*

PP2: *…we have bilateral, uhm, skills development programmes with those countries. Like Cuba…, Russia…, Taiwan. So, we select students, you know, yeah, to benefit from this, uh, bilateral, uh, skills development programmes but, uh, there are those like for example in Ukraine, you know most of those students in Ukraine, they just go there, yeah, and then when, they are there, yeah, of course we do ascertain if the quality of education they are getting is up to standard… If like they are best of our minds (sic) and they want to go and pursue their training, in those countries, and we think they qualify and the universities they are training at are good enough, then we support them wherever they are.*


#### 3.1.9. How Are the Schemes Funded and What Do They Pay for?

Figure 3 shows a summary of the sources of funding for these schemes. In Lesotho, all funding is considered to be from the government regardless of the original source [32,33]. Both Eswatini’s Ministry of Labour and Lesotho’s NMDS rely on government taxes and loan repayments from beneficiaries for the sustainability of their schemes. Botswana’s schemes largely rely on government taxes. The Ministry of Public Service in Eswatini funds its schemes from government taxes and donor schemes such as the Commonwealth. These schemes fund for tuition (including stationery and instruments (e.g., stethoscope, etc.)), accommodation and living allowances and transport from the country’s capital city (Gaborone, Mbabane or Maseru) to the destination, if outside the country. Medical insurance and VISA costs are also covered for those outside the countries [4,29,31,32,33,34,43]. 

#### 3.1.10. What Are the Benefits for the Individuals?

Though not guaranteed a job after the completion of their undergraduate studies, funded individuals have better prospects of being employed than other graduates. Those enrolled through the in-service programmes of the three governments are guaranteed their jobs upon their return, and they also receive a salary during their studies. In Botswana, in-service beneficiaries would be given their full salary in their first year of absence and a half salary beyond that, but that has since changed:

GBL1: *So, Government decided: ‘ok, fine, uh, anybody who goes on government sponsorship, they will be given full salary irrespective of the number of years that you go for’.*

In Eswatini, they still implement a staggered salary where the salary of in-service beneficiaries reduces with each year that the employee is away for [31]. These employee beneficiaries receive a full salary in the first year, 75% in the second year, 50% in the third year and 25% from the fourth year onwards. This then leaves beneficiaries with a 75% shortfall in income from the third year of study. Even though policymakers say this is implemented for fairness, some beneficiaries previously found this to be unfair, as it impacted their pre-enrolment bills, which continue through their studies [31].

PP1: *Uh, you can’t be getting your full salary whilst you are not doing anything in terms of work. It wouldn’t be fair to those who are at work.*

Lesotho applies a similar principle where the salary is halved from the first year of absence.

### 3.2. Understanding the Implementation and Implementers of Return-of-Service Schemes in the Three Countries (Process and Actors)

#### 3.2.1. How Are the Service Needs Determined?

Service needs and the number of beneficiaries to be funded per programme are determined through a consultative process with Ministries. For instance, prior to 2015, the government of Eswatini identified priority training needs, and training in health was amongst those identified to be high priority. The Human Resource Development Council (HRDC) within Botswana’s Ministry of Education provides a list of programmes that are supposedly scarce on an annual basis. The rationale is that the supported programme must be able to help sustain the students by getting employment in the economy after the completion of studies. The processes are similar for Lesotho’s undergraduate beneficiaries, as explained by respondents:

FZ1: *So, that’s why we are guided by… HRDC, they call it top occupants’ fields.*

DS1: *After finalisation of the priority areas, we publish them for the people to know that these are the programmes that we will be taking out of the country and then when it’s time for applications, we issue a notice advert that, applications are now open for these particular programmes and these are the conditions and now you apply… our budget will determine how many students we take because of the issue of affordability. If we can only afford about 200, then we’ll take 200 if we can afford up to 400, then we’ll take 400.*

In Botswana, units and health facilities are asked to provide training needs for the next financial year based on their identified priority service needs. These needs are often informed by the service pressures and employee performance development plans/reviews. These would then be sent to the National Ministry of Health’s human resources development unit for consolidation and alignment with the available training budget. The final list of priority areas is based on the available budget. Similar processes are undertaken in Eswatini (at inter-ministerial levels for public servants) and Lesotho. As explained by one participant:

DS3: *…within that limit, they do not come to us and prescribe how many we can have, but they will say: ‘Ministry of Health I only have this amount of money… so, what are your priorities?’, and then we would stipulate our priorities… then we would say within the one that allows us to recruit new people, our priorities will be doctors, nurses and what…*

#### 3.2.2. Application Process

The Botswana Ministry of Education releases an advertisement each year around February, when matric results come out. In Lesotho and Eswatini, applications start in August/September of the year preceding applications. For in-service training policies, the advertisement for positions follows the identification of priority training areas. These positions are advertised on various platforms, including notice boards at workplaces, social media, print and electronic media and word of mouth.

DS1: *Our Facebook page is more effective. We also use the state radio…and any other platforms that are available. For example, I just give it to you through WhatsApp, you issue it to the, uh, WhatsApp groups… We also take it to the Ministry website. So, it’s actually more or less like word of mouth…and we also put it on our notices.*


#### 3.2.3. How Are Eligible Beneficiaries Selected from Applicants?

In all the countries, a Board or Committee will assess the applicants and decide. In Lesotho, the final list is signed off by the Minister responsible for NMDS (Minister of Development and Planning). These examples from two participants highlights the process:

FZ1: *It will be a Board where somebody comes and present to say we have these 300 students looking for sponsorship and then we look at the applications and then we make recommendations as a Board.*

DS1: *…we first establish a team that’s going to work through all the applications from the start to finish. … then the selection is taken. We have a Council that… that oversees, the whole NMDS [National Manpower Development Secretariat] operations. So…, after capturing of the candidates they are then forwarded to, our Council… and the Council will make the… decision on the selected and who is not. From the Council then it will go to our Minister, but our Minister is just to show him that ok, these are the students who have been selected, these are those that have not been selected. Then the Minister, when he satisfied with what he has gotten then he will approve.*

#### 3.2.4. How Are Beneficiaries Monitored during Their Studies? 

Eswatini and Lesotho do not have formal ways of monitoring beneficiaries during their studies other than the expectation for beneficiaries to submit their results annually.

DS1: *We don’t have such a system, that is why people are defaulting.*

The Botswana Ministry of Education, on the other hand, relies on their consular staff in the foreign countries. No specific monitoring is carried out for those studying within the country.

FZ1: *We have Education outages who actually are staff members from the Department who are stationed in missions outside the country. So, their job is to monitor the students’ performance and welfare during their course of studies. But once the students have completed and they are back in Botswana, no, we don’t really make contact with them at all. Unless they come back to us for another sponsorship.*


#### 3.2.5. How Are Beneficiaries Recruited into Employment?

Whilst in-service beneficiaries have a straightforward arrangement in that they first return to the job they held before funding support whilst awaiting translation into the new role, this is not as clear for pre-service beneficiaries. In Botswana, the Ministry of Education requires health sciences beneficiaries to complete a certain form in the last month of their final year. In this form, beneficiaries enter their preferred placement hospitals. These forms are then sent to the Ministry of Health by the Ministry of Education so that they are aware of the number of completing students. This is the first and last formal communication between the two Ministries in this process of selection and placement. 

In Eswatini and Lesotho, undergraduate beneficiaries apply for a vacant position just like everybody else by approaching the Ministry of Health. In these two countries, health sciences beneficiaries are then expected to inform the Ministry of Labour or NMDS once they have started working so that repayment arrangements can be set up. As narrated by two respondents:

DS1: *So, the Ministry of Health will then place them, I don’t really know how that happens, but they register with the Ministry of Health, and the Ministry of Health will place them according to the vacancies, I think.*

DS3: *…what they normally do, is that they would apply to the Ministry of Health, they introduce themselves like, I am so, and so who is qualified recently, qualified somewhere, I am completing on this date. So, I am applying for a job…*

#### 3.2.6. How Are Beneficiaries Monitored after the Completion of Their Studies?

In the three countries, all the funding entities do not have mechanisms to monitor beneficiaries’ fulfilment of the service obligation, nor do they have mechanisms to trace health professionals who have defaulted or emigrated. Where there are funds expected to be repaid, debt collectors are often contracted to follow-up on beneficiaries who are behind on their payments. Eswatini’s in-service training policy is the only one that makes mention of a detailed monitoring and evaluation plan [31]. The monitoring and evaluation plan includes the monitoring of beneficiaries during their studies, feedback from beneficiaries on the relevance of training and the performance of beneficiaries post-completion of studies [31].

#### 3.2.7. Who Are the Actors Involved in the Success of These Schemes?

Figure 4 summarises the stakeholders that are directly or indirectly involved with the schemes. Lesotho’s policy mandates that the NMDS Director keep a record of each student to whom a bursary has been granted [32,33]. The Ministry of Health is a passive stakeholder for all pre-service schemes, as they are not involved with the selection or monitoring of beneficiaries during and after their studies. 

## 4. Discussion

This study of three Southern African countries found similarities in the design of their RoS schemes. Policies governing RoS schemes were all generic and not specific to the health sector. RoS schemes in these countries were introduced to address critical skills shortages, improve employability prospects for citizens, have competent public sector employees by global standards and aid the career pathing of government employees. All pre-service programmes were administered outside of the Ministry of Health. Whilst both Eswatini and Lesotho’s in-service schemes are administered outside the Ministry of Health, Botswana’s is administered within the Ministry of Health. Pre-service beneficiaries have both partial repayments and service obligations in Eswatini and Lesotho. Botswana’s undergraduate health sciences beneficiaries generally have no repayment obligations, as they mostly benefit from the top achievers’ programme, which only has a service obligation. The policies are all older than 20 years and have never been reviewed since inception. However, there are some prescripts that have been operationally amended over time due to changing practices, lessons from other countries and needs to ensure fairness. Pre-service beneficiaries are mostly selected based on academic merits, with little weight being put on the origins of the individual and their socio-economic status. Whilst most RoS literature have been from high-income countries (mostly the USA), besides several studies from South Africa, Malawi, Zambia and Sri-Lanka, there is limited literature from low- and middle-income countries in this area [14,21,44,45,46,47,48,49,50,51,52,53,54,55,56,57,58,59].

Intersectoral collaboration is one of the main strategies that was advocated for in the 1978 Alma-Ata declaration [60,61]. At the time, recognition was already made that health for all would only be attained if there were multisectoral actions [62]. The governance of RoS schemes in these three Southern African countries therefore presented a perfect opportunity for multiple Ministries to work together to improve the lives and livelihoods of their citizens. This research therefore provides valuable lessons on the dos and don’ts of collaborative governance, as it shows that multisectoral collaboration is a non-negotiable necessity in ensuring health systems’ efficiency and good health outcomes but is not sufficient by itself [62,63,64]. As Adeleye et al. [62] advised, success requires intentional, strategic and coordinated efforts of sectors.

As articulated by Eswatini’s Human Resources for Health strategic plan for 2012–2017, the coordination of RoS beneficiaries between the Ministry of Health and Ministry of Labour and the Ministry of Health and Ministry of Public Service (create posts) is weak [20]. Botswana’s 2007–2016 HRH planning strategy also noted that there were fewer health sciences beneficiaries than the target for each of the seven years of assessment (2000–2006) [18]. The conclusion reached was that this could have been poor knowledge of the schemes by potential beneficiaries [18]. As acknowledged by only Eswatini, but not unique to them, there are no obligatory mechanisms instituted to commit graduates to serve the public health sector and/or underserved areas [18,19,20]. Furthermore, all three countries have dedicated HRH planners, but it is not clear how their processes feed into the country agreements that guide RoS scheme implementation [18,19,20]. Eswatini acknowledged the lack of HRH planning capacity in the country [20]. It is therefore important for a policy to be explicit about its intentions, the outcomes that will be minimally acceptable for its success and how it will be evaluated [65].

It is commendable that the policymakers have a fairly good idea of the origins of the schemes. The fact that the policies have not changed over many years makes it easier to preserve institutional memory and transfer from one generation of administrators to the next. However, it is important to recognise that policies are only as relevant as the existing laws, trends, practices and technologies of the day; if any of those change, then they need to be reviewed [2,63]. This is evidenced by the recent equalisation in the treatment of male and female in-service beneficiaries in Botswana, where the policy discriminated against pregnant women [29]. Upon recognising this, stakeholders were to then get together and formally review the policy document following their pre-defined procedures. Consistent with the principles of a good policy that suggest review timelines in a policy [65], a policy document has to therefore have pre-determined review standard operating procedures and timelines, e.g., every 3 or 5 years.

Rural origin health sciences students are said to also be three times more likely to return to their places of origin than their urban counterparts [48,52,66,67,68,69]. The fact that academic merit is the mainstay of these schemes therefore ignores this overwhelming evidence [48,52,66,67,68,69]. Furthermore, the Ministries of Health are passive role players in the implementation of RoS schemes. Governments have to also make an effort to increase the internal training capacity, which in itself requires an investment in the training of a crop of internal educators.

This study notes six major challenges and provides some suggestions for improvement. First, the results of this study suggest it is unlikely that the RoS planning process follows the evidence guidance provided by Mabunda et al. [22] and Sousa et al. [2]. These authors [2,22] propose that HRH planning should be informed by a scientifically determined need (e.g., burden of diseases), the skills-mix that will best meet the needs should be determined, a beneficiary should be matched against a pre-identified service area, the efficiency should be improved to minimise waste and attrition and future salaries that will be used to pay beneficiaries should be considered. Second, the policies do not all have an evaluation plan. Third, the policies do not have room for incorporating principles of the WHO guidelines on HRH policy development since they are dated [24,25]. Frameworks help with the standardisation and simplification of complex processes. Fourth, the fact that the employment of these beneficiaries is not guaranteed, and it is left to the beneficiary to find employment leaves room for frustration and/or exploration of other employment options and thus the inefficiency of the programme through the leakage of trained health professionals. Fifth, coordination between the Ministry of Health, specifically the Human Resource Management arm, and the implementing unit and/or Ministry has to be strong to improve the monitoring of the schemes throughout the implementation cycle. Sixth and last, considering the fact that governments have opted for an approach where RoS policies apply transversally between Ministries, they need to have clearly defined and specific roles for each of the Ministries receiving the beneficiaries.

This study is limited by poor archiving in all the countries, as all implementing organs felt that there could at least be one more policy document that could provide clarity on the origins of the schemes but that could not be located. However, this was supplemented by rich information acquired from the participants and documents. Even though the Eswatini Ministry of Health did not have a participant, inputs acquired from the Ministry of Labour and Ministry of Public Service provide a good description on the origins and implementation of RoS schemes, therefore fulfilling the study aims.

## 5. Conclusions and Recommendations

Bursaries, scholarships and grant-loan schemes play an important role in upskilling health professionals, creating employability prospects for the youth in the health sector, benchmarking skills and improving the networking potential for beneficiaries as they study in neighbouring countries and abroad. Even though the precise origins of the schemes are not known, there are fairly good estimates on when the schemes began and how they have evolved over time. The efficiency of these schemes is further impeded by poor coordination between the implementing/or administering agency, e.g., the Ministry of Education and the Ministry of Health. This therefore requires a strategy for the creation of cooperation and coordination plans between the Ministry of Health (specifically, human resource management) and the administering agency to ensure that the potential for RoS schemes to provide a solution to the human resources for health gaps is optimised. It is therefore recommended that the implementation of these schemes be guided by evidence-based decision making and schemes allocated following a needs-based analysis. These require an internal capacity on epidemiology, health economics and HRH planning. In addition, RoS information systems must be interoperable with other HRH information systems, including relevant national databases, e.g., home affairs registry, taxation system, etc.

## Figures and Tables

**Figure 1 healthcare-11-01512-f001:**
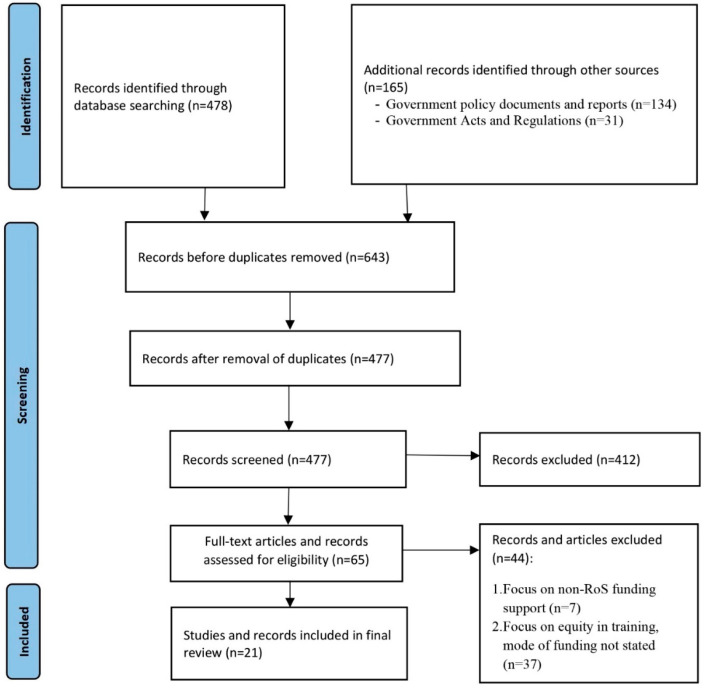
Flow chart of the literature and document review search.

**Figure 2 healthcare-11-01512-f002:**
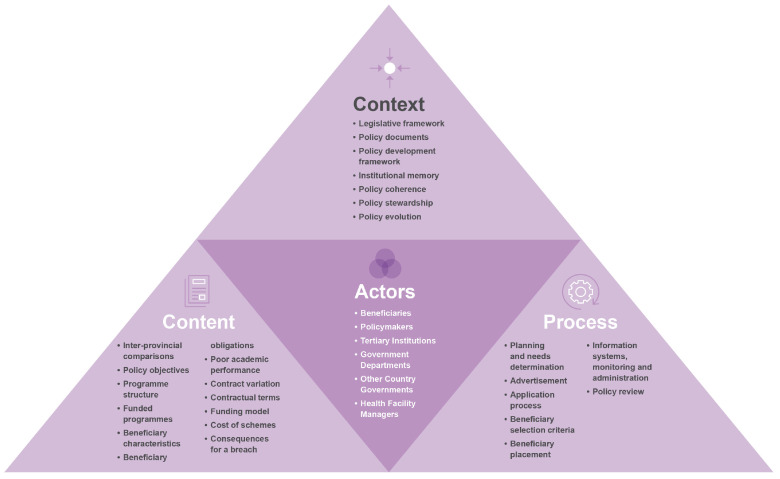
Walt and Gilson policy framework used to triangulate the data.

**Figure 3 healthcare-11-01512-f003:**
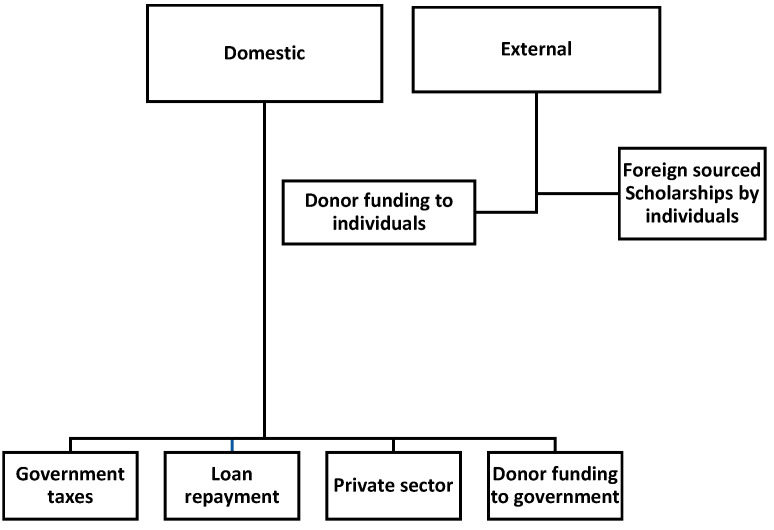
Sources of funds for Ros schemes.

**Figure 4 healthcare-11-01512-f004:**
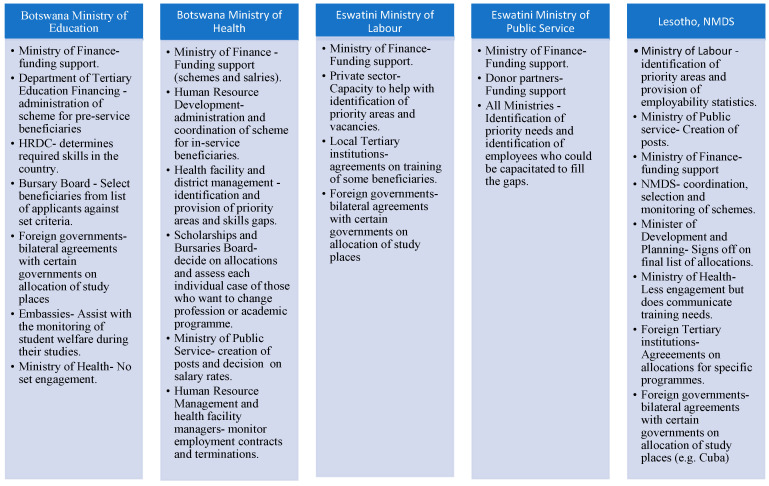
Summary of actors involved with RoS schemes in the three countries.

**Table 1 healthcare-11-01512-t001:** Summary of country indicators for Botswana, Eswatini and Lesotho [6,10,11,15,16].

Indicator	Botswana	Eswatini	Lesotho
Population, 2022	2,441,162	1,184,817	2,175,699
Population density (per square kilometre), 2022	4.2	68.2	71.7
GDP per capita (USD, Billions), 2021	7347.6	4214.9	1166.5
Human development index rank, 2022	0.7	0.6	0.5
Income level, 2020	Upper-middle	Lower-middle	Lower-middle
Doctor density (per 10,000 population), 2018	3.8	2.5	4.7
Nursing and midwifery personnel density (per 10,000 population), 2018	37.7	41.4	32.6
Life expectancy (years), 2020	69.9	61.1	55.7
Under-5 mortality rate (per 1000 live births), 2020	44.8	46.6	89.5
HIV/AIDS prevalence (%), 2019	20.1	27.7	24.8
Tuberculosis incidence (per 100,000), 2020	236	650	319

**Table 2 healthcare-11-01512-t002:** Scheme origins and configuration.

Pre-Service Programme			
Characteristic	Botswana	Eswatini	Lesotho
Administering Ministry/Agency	Education	Labour	NMDS
Earliest policy found	1995	1977	1978
Countries of study	- Locally (mostly nursing, paramedics, some medical students, pharmacy and some allied health professions) - Internationally (South Africa, Cuba, Russia, UK, Australia, Iran, USA, etc.)	- Locally (mostly nursing and paramedics) - Internationally (South Africa, Zambia, Tanzania, Ukraine, Taiwan, Russia, etc.)	- Locally (mostly nursing, paramedics, pharmacy, environmental health and nutrition) - Internationally (South Africa, Zambia, Tanzania, Nigeria, Algeria, China, etc.)
**In-Service Programme**			
Administering Ministry/Agency	Health	Public Service	NMDS
Earliest policy found	1996	2000	1978
Countries of study	- Locally (some undergraduate programmes, some nursing and medical specialisation, e.g., family medicine, etc.) - Internationally (South Africa, UK, Australia, Iran, USA, etc.)	- Locally (mostly nursing and paramedics) - Internationally (South Africa, Zambia, Tanzania, Ukraine, Taiwan, Russia, etc.)	- Locally (mostly nursing and pharmacy) - Internationally (South Africa, and any country in the world the beneficiary chooses)

NMDS = National Manpower Development Secretariat.

**Table 3 healthcare-11-01512-t003:** Beneficiary characteristics and obligations in the three countries and the different responsible authorities.

Characteristic	Botswana		Eswatini		Lesotho
	Education	Health	Labour	Public Service	National Manpower Development Secretariat
Suitability criteria	- Citizens (children of Diplomats are considered, citizens based in foreign countries are considered equally as those based locally) - High academic grades	- Educational suitability - Botswana citizen - Age limit - Physical fitness - Minimum of 2 years of pensionable service in the government - Seniority assessed together with need - In line with current work	- Swati citizen - Must be applying for priority programme in that year - ASP points above a particular threshold - With motivation, being of low socio-economic status	- Swati citizens - Permanent (not on probation) and pensionable government employee - Be in their Ministry’s training plan for that financial year - Must be applying for a priority programme - Must have an elaborate re-integration plan - Must not be in possession of the same level of qualification	- Score applicants based on: - Age (prefer younger applicants (under 30) for undergraduate qualifications) - Previous qualification (if present) and relevance to current application - Grades in previous qualification (if present): those with merit will receive more points than a person who just has a pass - Repayment of previous sponsorship (if present)
Beneficiary obligations	- Top achievers and health sciences beneficiaries do not have a repayment obligation, only service to the government - Beneficiaries on grant-loan schemes are not forced to work for the government	- Pass - Return to government for service obligation	- Study programme as per agreement - Pass - Inform Ministry that studies have been completed - Return to the economy	- Return to original post after the completion of studies	- Stay enrolled in programme agreed on - Good academic performance (to pass) - Share results with unit annually - Sign contract annexure annually - Return to the country upon the completion of studies - Repay at least 50% of funding (within 5 years) if in the public sector
Service period	Duration funded ×1	- 1–12 months of study = 1 year - >12 months = duration of study funding (rounded off to nearest year) +1 year, e.g., 18 months = 3 years’ service Maximum service period = 5 years	Duration funded ×2	Duration funded +1	Duration funded ×2 (minimum service period = 3 years if funded for 1 year)
Repayment of funds	None for health science beneficiaries (100% repayment for non-health sciences beneficiaries)	None	- 50% repayment if in the country - 100% repayment if outside the country	None	- 50% repayment if in the country’s public sector - 65% repayment if in the country’s private sector - 100% repayment if outside the country

## Data Availability

The data used in this study are available from this article.

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
