# Peer review of "How Were Return-of-Service Schemes Developed and Implemented in Botswana, Eswatini and Lesotho?"

_healthcare, 2023, doi:10.3390/healthcare11101512_

Round 1

Author Response

Dear Editor:

Thank you for reviewing our manuscript titled: Running return of service schemes in Southern Africa: lessons from Botswana, Eswatini and Lesotho.

Comments from the reviewers have helped enhance our article and we have addresses all comments in chronological order, stratified by reviewer.

Reviewer 1

General Comments

Comment: Paper by Sikhumbuzo A.Mabunda and His team, "Running return of service schemes in Southern Africa: lessons from Botswana, Eswatini and Lesotho", has eighteen pages and is divided into a few parts dealing with different aspects of the subject being investigated.

Response: Noted.

Comment: It is interesting work that can play an important role in evaluating the return of service in respective countries.

Response: Noted, thank you.

Comment: But on the other hand the paper has unclear structure even though the titles look rather typical.

Response: Thank you for the comment. Our intention was to triangulate data from three sources to explain the historical development of return-of-service schemes in these three countries and how the current practices have been informed by their formation, and how they have evolved over time. Line 78-99 of the Methods section and line 143-153. We have further added a statement in the results (line 1501-153) to help clarify the confusion: “There was good alignment between the three data sources, as such information is presented under a single theme, marked by citations if extracted from literature or policies and direct statements are in italic font”. 

Comment: One of the most important features of scientific papers is clarity of thought and style. Besides the scientific part of the issue also the presentation of the whole problem has the same, or almost the same, value. The reason for taking scientific literature in hand is enlargement and upgrade the knowledge to extend it into public health. Therefore such knowledge should be available and accessible. Here, in the sense of making it understandable. But before the final decision regarding the time of publication, some changes in the text are inevitable. Because first and, at the end, the last impression after reading this paper is difficult to read due to complicated structure, unclear presented ideas, and other specific doubts making together feeling of mess. So, the general tip is to thoroughly overview the whole paper ;the sort of inspection to find every piece of the paper that could be obstacles for fluent and logical presentation of the subject.

Response: Thank you for the suggestion. The article has been comprehensively reviewed to improve the flow and the title revised to link with our objectives.

Comment: The background informations are already in the text. So, the many doubts are more of form than of ideas. Beneath some propositions and suggestions. As the examples making the idea clear.

Response: Noted.

Comment: In the title, the words "lessons"are not necessary. The subject of the study is not the preparation of the lessons but the description of the design and implementation of Ro's schemes in the three countries and adherence to the WHO principles.

Response: Thank you for the comment, the title has now been revised to: How were return-of-service schemes developed and implemented in Botswana, Eswatini and Lesotho?

Comment: If this aspect of the study is so crucial for the authors that the proposed change is hard to introduce, the question of reshaping the whole text should be considered. It is necessary to check reference papers.

Response: Thank you for the comment, the title has been revised.

Comment: In a few places, they are not relevant to the subject. A typical example is taking Alma-Ata Declaration as the justification for intersectoral collaboration. This is out of the question that the Alma-Ata Declaration of 1978 is the milestone of the twentieth century in global health. The Declaration identified primary health care as the key to the Health for All idea. The Declaration is still relevant for organizing healthcare systems because its principles are critical for the contemporary approach to such ideas as the UHC concept. But to call The Declaration the basis for the justification of intersectoral collaboration to improve the return of service schemes?

Response: Thank for comment. Our aim of bringing in the Alma-ata declaration is because, the declaration aimed to ensure attainment of health for all through the alignment of whole of government strategies and purpose for this goal. One of the reasons for failure to retain health professionals in rural areas is for instance, the lack of educational facilities for children, the lack of jobs for spouses, poor infrastructure, etc. If RoS schemes are therefore implemented across Ministries, these challenges can therefore easily be dealt with through coordinated multi-Ministerial forums, something which is unfortunately not happening in these countries. We are, however, happy to make this point even clearer if our explanation doesn’t do so. 

Comment: Indeed, that is only one example of the problem. Authors indeed use a multimethod research design. It is also a good idea to put some information in annexes. But in many places, there is a lack of information, at least substantial, regarding the methods and the reason for using them. Again just as an example. The concept of Gill Walt and Lucy Gilson for analyzing both the policy document review and literature. Splendid. But their ideas are more than 30 years of age. Are they still valid? Globally or only in some countries.?

Response: Thank you for the comment. A figure has been added to demonstrate how the Walt-Gilson policy framework was used. This framework as much as it’s been in existence for decades is still relevant to this day as it makes it easier for the triangulation and structuring of information.

Comment: There are three sources of information. Namely, literature review, political review and interviews of experts. The interrelation between them is not clear. Part 3 of the paper should be reshaped, making it more transparent. A good starting point to introduce changes is to answer why the text is divided into parts by titles in small italic letters and other places by bold words.

Response: Thank you for the comment. Italics refer to direct quotes of participants with their pseudonym in Bold (a random ID). Other bolded text is reserved for headings or sub-headings in Tables and text. Policy reviews and literature reviews are referenced.  

Comment: The names of professions differ in different places: physicians, doctors, nurses, midwives, health professionals, generalist nurses etc. It is not clear enough to understand the reason for that and needs explanations.  

Response: Thank you for the comment. Physicians and doctors have been standardised to doctors in text from literature or the researchers’ own interpretation. In addition, in some contexts, nurses are trained as generalists before undertaking midwifery as a specialty and in other contexts midwifery is a basic training, so the distinction is there to acknowledge these contextual differences. Generalist nurses have therefore been changed to non-specialist nurses. Health professionals refers to any category of the above (mostly in combination) including pharmacy, dentistry, etc. 

Comment: The paper is in some parts, qualitative. But so few investigated people and the subsequent impact on results have to be interpreted. There are only a few examples to make clear my doubts. And at the end I have to underline the necessity to overview the whole text to make it reasonable and readable, to carry on clear lecture and prepare introduction for changes in the subject into consideration.

Response: The qualitative interviews were intentional and only targeted policymakers who manage RoS schemes and they are few in nature (e.g., a country will have one president, etc.). This article managed to reach 90% of the policymakers who are influential in the setting up of these schemes. Only the Ministry of Health in Eswatini (who do not deal with RoS) could not be interviewed as they didn’t view their role as being important.

Reviewer 2 Report

Thank you for sending me to review the interesting and comprehensive paper titled "Running return of service schemes in Southern Africa." I find the paper relevant to countries where there is a shortage of medical professionals and a massive migration of professionals to richer countries to improve their living conditions.

I have a few minor comments:

1. It is recommended to organize the results section to clarify what emerged from the interviews and what was from the document/policy analysis.

2. In the discussion, referring to what is happening in other countries in this field is recommended.

3. What are these countries doing to return professionals who emigrated? Are there any such governmental programs?

4. Recommendations for future studies in the field should be added.

Author Response

Dear Editor:

Thank you for reviewing our manuscript titled: Running return of service schemes in Southern Africa: lessons from Botswana, Eswatini and Lesotho.

Comments from the reviewers have helped enhance our article and we have addresses all comments in chronological order, stratified by reviewer.

Reviewer 2

General comment

Comment: Thank you for sending me to review the interesting and comprehensive paper titled "Running return of service schemes in Southern Africa." I find the paper relevant to countries where there is a shortage of medical professionals and a massive migration of professionals to richer countries to improve their living conditions.

Response: Noted, thank you.

Specific comments

I have a few minor comments:

Comment: 1. It is recommended to organize the results section to clarify what emerged from the interviews and what was from the document/policy analysis.

Response: The unique feature of this study is the triangulation of three different sources of information. To maintain this, the qualitative interviews are reflected by direct quotes or a reference to policymakers’ assertions whilst the policy and literature reviews are reflected by the presence of in-text citations. We have also added Figure 1 to reflect the features that were triangulated using the framework. In addition, line 151-153 of the results has added a statement to state an easy way of separating the text: “There was good alignment between the three data sources, as such information is presented under a single theme, marked by citations if extracted from literature or policies and direct statements are in italic font”.

Comment: 2. In the discussion, referring to what is happening in other countries in this field is recommended.

Response: Thank you for the comment and input. This statement has been added in the discussion: “Whilst most RoS literature has been from high income countries (mostly the USA), besides several studies from South Africa, Malawi, Zambia and Sri-Lanka, there is limited literature from low- and middle-income countries in this area” in lines 514-517. In addition, lines 554-555, and 561-563 also quote other countries’ experiences.

Comment: 3. What are these countries doing to return professionals who emigrated? Are there any such governmental programs?

Response: Thank you for the comment. Line 484 in the Results has appended the statement: “… nor do they have mechanisms to trace health professionals who have defaulted or emigrated”.  

Comment: 4. Recommendations for future studies in the field should be added.

Response: Recommendations have been added in line 598-603: “It is therefore recommended that implementation of these scheme be guided by evidence-based decision making and schemes allocated following a needs-based analysis. These require internal capacity on epidemiology, health economics and HRH planning. In addition, RoS information systems must be interoperable with other HRH information systems including relevant national databases, e.g., home affairs registry, taxation system, etc”.